# Effectiveness of Cognitive Behavioral Therapy (CBT) on Psychological Distress among Mothers of Children with Autism Spectrum Disorder: The Role of Problem-Solving Appraisal

**DOI:** 10.3390/bs14010046

**Published:** 2024-01-10

**Authors:** Enas Mahrous Abdelaziz, Nourah Alsadaan, Mohammed Alqahtani, Nadia Bassuoni Elsharkawy, Marwa Mohamed Ahmed Ouda, Osama Mohamed Elsayed Ramadan, Mostafa Shaban, Evon S. Shokre

**Affiliations:** 1College of Nursing, Jouf University, Sakaka 72388, Saudi Arabia; emabdelhamid@ju.edu.sa (E.M.A.); nelsharkawy@ju.edu.sa (N.B.E.); mmouda@ju.edu.sa (M.M.A.O.); omramadan@ju.edu.sa (O.M.E.R.); mskandil@ju.edu.sa (M.S.); 2Psychiatric Mental Health Nursing Department, Faculty of Nursing, Cairo University, Cairo 11562, Egypt; 3Department of Nursing, College of Applied Medical Sciences, King Faisal University, Al Hofuf 31982, Saudi Arabia; mealqahtani@kfu.edu.sa; 4Maternal and Newborn Health Nursing Department, Faculty of Nursing, Cairo University, Cairo 11562, Egypt; 5Pediatric Nursing Department, Faculty of Nursing, Damanhur University, Damanhur 22516, Egypt; 6Pediatric Nursing Department, Faculty of Nursing, Cairo University, Cairo 11562, Egypt; 7Geriatric Nursing Department, Faculty of Nursing, Cairo University, Cairo 11562, Egypt; 8Psychiatric Mental Health Nursing Department, Faculty of Nursing, Fayum University, Fayum 63514, Egypt; ess03@fayoum.edu.eg

**Keywords:** autism spectrum disorder, cognitive behavioral therapy, problem-solving, psychological distress, randomized controlled trial

## Abstract

Mothers of children with autism spectrum disorder (ASD) face considerable psychological distress. Cognitive behavioral therapy (CBT) has shown promise in reducing caregiver burden, but evidence in the Arab world is lacking. Problem-solving appraisal training may optimize CBT’s benefits. This study evaluated a tailored CBT program emphasizing the role of problem-solving appraisal in managing distress through the process of CBT. Sixty mothers were randomly allocated in a 1:1 ratio to either the CBT intervention group or the control group. The CBT group received 3-month sessions incorporating problem-solving appraisal training within a CBT curriculum from October 2022 to April 2023. Problem-solving techniques are focused on equipping individuals with the skills to identify, approach, and effectively resolve problems, leading to a reduction in stress levels and an improved capacity to cope with difficult situations. All mothers completed assessments of socio-demographics such as age, level of education, marital status, occupation, and adequacy of family income, the Depression Anxiety Stress Scale (DASS-21) and the Problem-Solving Inventory (PSI) before and after the program. The CBT group showed significant pre-to-post reductions in depression, anxiety, stress, and general psychological distress compared to controls (all *p* < 0.05). They also exhibited significant improvements in problem-solving confidence, approach-avoidance style and personal control (all *p* < 0.05). The customized CBT program markedly improved psychological well-being for mothers of children with ASD in Egypt. Incorporating problem-solving techniques may optimize CBT’s benefits cross-culturally. These findings have implications for the implementation of evidence-based support programs for families impacted by ASD worldwide.

## 1. Introduction

Autism Spectrum Disorder (ASD) is a complex neurodevelopmental condition characterized by challenges in social interaction, communication, and behavior, that persist throughout an individual’s lifetime [1,2,3,4]. Its prevalence is significant, with the Centers for Disease Control and Prevention (CDC, 2020) estimating that approximately one in 68 children worldwide are diagnosed with ASD [5]. The condition occurs more frequently in men, with a male-to-female ratio of about 3:1. The complexity and nature of ASD mean that it manifests itself in a range of severity, impacting individuals and their families differently [6,7].

The challenges posed by ASD extend beyond the individual to their immediate caregivers, particularly mothers [8,9]. These caregivers often face considerable psychological distress due to the chronic nature of the disorder and the severity of symptoms [10,11]. This distress is compounded by broader societal issues, including stigma, lack of acceptance, and inadequate support services [11,12,13,14,15]. Studies by [16,17] highlight these stressors and challenges, emphasizing the need for effective management.

The psychological impact on mothers of children with ASD is profound [18]. They frequently encounter stressors such as managing their child’s behaviors, navigating social stigma, and grappling with the chronicity of the disorder [19,20,21,22]. This ongoing stress can lead to elevated levels of anxiety, depression, and overall psychological stress [23,24]. The literature, including works by [25,26], has consistently shown that these mothers are at a higher risk of psychological distress, which can also adversely affect their children’s development and well-being. In light of these challenges, there has been a growing emphasis on identifying effective interventions to support these caregivers [27,28,29]. Among the various therapeutic approaches, cognitive behavioral therapy (CBT) has shown considerable promise [30,31]. CBT is a structured, problem-focused method that has been effective in reducing emotional strain. It combines cognitive and behavioral techniques to help people become aware of and change negative thought patterns and behaviors, thus reducing stress and improving emotional well-being [32].

Despite the proven efficacy of CBT in various contexts, there remains a significant gap in the literature, particularly regarding its application among mothers of children with ASD in the Arab world [33,34,35,36,37]. While studies in Western and Eastern contexts, such as those of [33,38,39], have validated the usefulness of CBT in teaching problem-solving skills to this demographic, similar research is scarce in Arab countries. This gap is notable considering the cultural differences that may influence the stressors experienced by these mothers and the effectiveness of interventions such as CBT [40,41,42].

Furthermore, within the scope of CBT, problem-solving techniques have been identified as particularly beneficial. These techniques involve training individuals to identify, approach, and resolve problems more effectively, thereby reducing stress and improving their ability to cope with challenging situations [43,44,45,46,47]. Bandura’s (1986) thesis on self-efficacy and Beck’s (2003) work on cognitive therapy underscore the importance of these techniques in enhancing problem-solving abilities and consequently reducing psychological distress, particularly depression [48,49].

The role of community mental health nurses in this context is crucial [18]. They are often the first line of support for mothers of children with ASD, providing guidance, training, and emotional support [50,51]. These professionals can play an essential role in the implementation of CBT and in the teaching of problem-solving techniques [52]. However, there is a lack of structured programs and research that focuses on the training provided by these nurses in the Arab world, further highlighting the research gap this study aims to address [50,51,52,53,54,55,56]. Given the above considerations, this study aims to evaluate the effectiveness of a customized CBT program, emphasizing problem-solving techniques, to improve the psychological well-being of mothers of children with ASD in the Arab world. It seeks to fill the identified research gap by providing empirical evidence on the efficacy of such a program in this specific cultural context. The study hypothesizes that participation in the CBT program will lead to a reduction in psychological distress among these mothers, enhancing their problem-solving abilities and overall psychological well-being.

In conclusion, the need for research in this area is clear. With increasing awareness of ASD and its impact on families, particularly in non-Western contexts, it is imperative to explore and validate effective interventions. This study aims to contribute to this effort, providing new insights and potentially guiding future interventions to support mothers of children with ASD, thus improving the quality of life for caregivers and their children.

## 2. Materials and Methods

### 2.1. Research Question

“What is the efficacy of a tailored Cognitive Behavioral Therapy (CBT) program that incorporates problem-solving skills training to improve psychological well-being among mothers of children with autism spectrum disorder (ASD) compared to mothers who do not receive the intervention?”

### 2.2. Research Hypothesis

In light of the substantial relationship between problem-solving techniques, psychological distress, and CBT in individuals, the present study aimed to examine two hypotheses.

**H1.** *There are statistically significant differences, at the α level of 0.05, between the mean scores of the study and control groups on the posttest of total Depression Anxiety Stress Scale (DASS-21) and its sub-scale*.

**H2.** *There are statistically significant differences, at α level of 0.05, between the mean scores of the study and control groups on the posttest of total Problem-Solving Inventory (PST) and its sub-scale*.

### 2.3. Design

This study utilized a randomized controlled trial (RCT) design with pre-post assessments to evaluate the efficacy of the CBT intervention. Mothers who met eligibility criteria were randomly assigned to the CBT intervention group or a control group in a 1:1 ratio. The CBT intervention group received 3 months of weekly sessions of a customized CBT program that incorporates training in problem-solving skills. The mothers in the control group met with the researchers monthly for general discussion and needs assessment but did not receive CBT or other psychological interventions during the study period.

Both groups completed pre-intervention assessments of psychological distress (DASS-21 questionnaire) and problem-solving abilities (PSI inventory). After the 3-month CBT program concluded, all participants again completed the DASS-21 and PSI as post-intervention measures. This RCT design allowed for comparisons between groups, evaluating changes in outcome variables from baseline to post-intervention. It was aimed at evaluating the efficacy of the structured CBT program in improving psychological well-being and problem-solving skills relative to the control group who were not receiving CBT. Design and procedures were controlled for nonspecific therapeutic effects and potential confounding.

### 2.4. Settings

The study was conducted at three governmental educational facilities in El Beheira governorate, Egypt: Itay El Barud, Kafr El Dawwar, and Abu Hummus. These sites were selected as they contain clinics serving children with autism spectrum disorder (ASD) and other special needs. Participants were recruited from mothers accompanying their children for services at these clinics.

### 2.5. Eligibility Criteria

#### 2.5.1. Inclusion Criteria

Mothers aged 20–50 years.Mothers must be the primary caregiver and resident of a child diagnosed with autism spectrum disorder (ASD) for 2 years.The child’s diagnosis of ASD must be made by a psychiatrist.Mothers can have only one child diagnosed with ASD.No prior participation of mother or child in rehabilitation or intervention programs.

#### 2.5.2. Exclusion Criteria for Mothers

Current or past psychiatric diagnosis.Attendance of <5 CBT sessions.Severe levels of psychological distress (scores in severe ranges on DASS-21 subscales).

#### 2.5.3. Exclusion Criteria for Children

Estimated intellectual disability (IQ < 70).Severe symptom deterioration or regression.Ongoing treatment with psychiatric medications.Comorbid diagnoses of multiple developmental disorders.

### 2.6. Participants and Sample Size Determination

A total of 206 mothers caring for a child with ASD were screened for eligibility. Of these, 67 met all inclusion criteria and were informed of the study purpose. To determine the appropriate sample size, power analysis was conducted using G*Power 3.1 based on medium expected effect sizes from similar prior studies. Target enrollment was set at 60 participants total (30 per study arm) to provide 80% power for detecting significant differences in the outcome measures. Seven mothers participated in a pilot study to refine the procedures. The remaining 60 eligible mothers were randomly assigned to either the CBT intervention or control condition at a 1:1 ratio using computer-generated random number sequences. A statistician uninvolved with enrollment prepared the allocation sequence. Participants and investigators were blinded to group assignments until after baseline assessments. All participants completed questionnaires assessing psychological distress (DASS-21) and problem-solving skills (PSI) at both pre-intervention baseline and again post-intervention. The final analyzed sample included 30 mothers in the CBT group and 30 controls [35,36,37].

### 2.7. Data Collection Tools

This study utilized three validated self-report questionnaires:Sociodemographic Questionnaire: Collected information on the age of mothers; level of education; marital status; occupation; the adequacy of family income; and the age, sex, and severity of autism of the child.Depression anxiety stress scale (DASS-21): A 21-item scale assessing symptoms of depression, anxiety, and stress over the past week [57]. The Arabic DASS-21 translated by [58] was used. It has robust reliability and validity in Arab populations [58]. In this study, internal consistency was high (Cronbach’s alpha = 0.903).Problem-Solving Inventory (PSI): A 32-item scale measuring perceived problem-solving abilities across three subscales: problem-solving confidence, approach-avoidance style, and personal control [59]. The PSI has strong psychometric properties and utilizes a 6-point Likert scale [60]. Lower scores indicate better perceived problem-solving [61]. In this study, PSI had high internal consistency (Cronbach’s alpha = 0.938).

All instruments were reviewed by experts to verify the validity of the face and content. The pilot testing established good test-retest reliability. DASS-21 and PSI aligned with the study aim to assess the effects of the intervention on psychological distress and problem-solving skills, respectively.

### 2.8. Ethical Approval

This study was registered prospectively on ClinicalTrials.gov (Identifier: NCT05568303), prior to participant recruitment, and following all policies for protocol registration. Ethical approval was obtained from the Research Ethics Committee for Health Affairs at the Faculty of Nursing, Damanhour University, Egypt (reference number 62) before initiating any study procedures. The ethics application outlined the study objectives, methodology, instruments, anticipated benefits and risks of participation, and data security provisions. Eligible participants were provided with detailed information about the study’s purpose, procedures, potential benefits and risks, their right to withdraw at any time, and confidentiality of personal data. Written informed consent was obtained from all participants prior to enrollment.

### 2.9. Procedure

#### 2.9.1. Study Randomization

Participants meeting the eligibility criteria were assigned unique identification numbers. A computerized random number generator was employed to randomly allocate participants into either the study or control group, maintaining a 1:1 allocation ratio from October 2022 to April 2023. This process ensured the randomness and unbiased nature of group assignment, crucial for the validity of a randomized controlled trial. Figure 1 provides a flow diagram detailing participant screening, enrollment, allocation, interventions, and analysis. Data were collected at baseline prior to intervention and again after the 3-month CBT program concluded. All procedures for recruitment, randomization, data collection, and analysis adhered to the registered protocol.

#### 2.9.2. Cognitive Behavioral Therapy Intervention

Thirty mothers from the El Behira governorate in Egypt, divided into four groups of 7–8 participants each, were selected for the study group. Over a three-month period, they received weekly CBT sessions lasting 60 min. However, at the end of each session, participants were given an additional 30 min to ask clarifying questions and ensure thorough mastery of the content taught. This resulted in a total session time of about 90 min.

These sessions focused on the utilization of problem-solving techniques within the cognitive behavioral framework to address psychological distress related to their children’s ASD.

The CBT curriculum was informed by the principles outlined by [62], integrating insights from previous research [59,62,63,64]. Key components included cognitive techniques to identify and alter the impact of automatic thoughts on emotions and behaviors and behavioral strategies such as activity scheduling, progressive task assignments, role-plays, and cognitive rehearsal. Special emphasis was placed on problem-solving techniques, involving steps to analyze problems, evaluate coping options, and formulate and implement effective plans.

Therapeutic methods within the CBT sessions included explanation, teaching, modeling, role play, reinforcement, feedback, and homework assignments. Group therapy strategies such as trust-building, empathy, acceptance, self-disclosure, and emotional catharsis [64] were also employed.

#### 2.9.3. Objectives and Teaching Methods

The primary objective was to enhance participants’ psychological well-being through cognitive and behavioral techniques, focusing on altering current thought patterns and behaviors and bolstering problem-solving skills. Methods such as brainstorming, lectures, discussions, examples, videos, role plays, and booklets were employed as teaching aids. Each session concluded with a summary, feedback, and clarification of any ambiguities.

#### 2.9.4. Phases of the CBT Program

The program was structured into three phases: assessment, implementation, and evaluation:-**Assessment Phase (two sessions):** Initiated with pre-testing procedures (T1), participants received comprehensive information about the study’s objectives and implementation strategies, both verbally and in writing. The study group received the program sessions across three hospitals simultaneously, always convening in the same room in outpatient clinics.-**Implementation Phase (Eight Sessions):** Each session combined theoretical and practical elements of CBT. The final segment of each session allowed for open discussion and inquiries. Session themes ranged from educating about ASD and its impact on family dynamics to teaching cognitive restructuring techniques and problem-solving skills.-**Sessions of the CBT program**-Session (1): During this session, participants received education about their child’s disease, including information about signs and symptoms, specific needs, principles of behavioral management skills, and the impact on their family. As part of their homework for this session, participants were instructed to document any symptoms exhibited by their child that contribute to challenges or difficulties.-Session (2): During this session, the researchers aimed to elucidate the connection between thoughts, emotions, and behaviors. They specifically addressed how biased thoughts can influence one’s emotions and actions. Participants engaged in discussions regarding the awareness of negative automatic thoughts (ATs) and the accompanying feelings that arise in stressful life situations. As part of their homework for this session, participants were assigned the task of reflecting on their understanding of emotions and examining their own emotional experiences.-Session (3): participants were encouraged by the researchers to take note of negative irrational thoughts, and discussions were held on how to effectively recognize these thoughts. Subsequently, participants were taught methods to transform and substitute negative thoughts with positive and logical ones, and they were also guided on how to reframe negative thinking patterns. Moreover, the session involved a detailed exploration of techniques such as guided imagery, meditation, and visualization.-Session (4): The researchers discussed the principles and techniques of cognitive restructuring, which involve replacing negative automatic thoughts (ATs) with more positive ones. In addition, the session addressed problem-based coping skills, the importance of developing positive friendships for effective problem-solving, and the use of modeling to manage stress. The researchers also assisted participants in practicing progressive relaxation techniques, including both active and passive approaches.-Session (5): During this session, participants were introduced to problem-solving appraisal techniques, which aimed to assist them in effectively dealing with various stressful situations. Additionally, the researchers highlighted the importance of self-instruction and self-talk as strategies to control negative thoughts. Participants were encouraged to explore and apply these techniques to manage their own cognitive processes and enhance their ability to cope with stress.-Session (6): In this session, participants were instructed in the step-by-step process of problem-solving using the solved technique. They were encouraged to visually illustrate their problems and select potential solutions. Through group discussions, participants had the opportunity to share their experiences and engage in open dialogue. Moreover, participants were prompted to express their inner feelings towards their problems, focusing on both positive and negative aspects.-Session (7): Participants were actively encouraged to follow the steps of problem-solving. The researchers emphasized the importance of systematically approaching problem-solving tasks and provided guidance to participants on each step of the process. By following these steps, participants were equipped with a structured approach to effectively tackle their problems.-Session (8): The researchers consolidated the key learnings from the previous sessions. Participants were given opportunities to practice the skills they had been taught, with a specific focus on their application in real-life situations. Emphasis was placed on utilizing these skills in various aspects of their lives. This phase aimed to reinforce the application of acquired skills and prepare participants for the challenges they may face beyond the therapy sessions.-**Evaluation Phase (two sessions):** Participants’ responses and homework assignments were used as indicators of session efficacy. Post-intervention assessments were conducted using the same instruments immediately after the program’s conclusion to evaluate the impact of problem-solving techniques in the CBT program.

#### 2.9.5. Program Validity

Prior to implementation, the program was reviewed and validated by three professors in psychiatric mental health nursing and two in pediatric nursing. This peer review process ensured that the program was academically sound and relevant to the target population.

### 2.10. Statistical Analysis

All statistical analyses were conducted using SPSS version 26.0 (IBM Corp., Armonk, NY, USA). Numerical data were presented as mean and standard deviation, while categorical data were presented as frequency and percentage. The Kolmogorov–Smirnov test was used to assess normality of data distribution. Comparisons between the two independent groups regarding categorical variables were performed using the Chi-square test or Monte Carlo test when more than 20% of cells had an expected count of less than 5. The Student *t*-test was used to compare normally distributed numerical variables between the two groups. The Mann–Whitney U test was used for non-normally distributed numerical variables. The paired *t*-test or Wilcoxon signed-rank test was conducted to assess changes from baseline to post-intervention in each group. Multivariate linear regression models were constructed to determine independent predictors of psychological distress and problem-solving skills while adjusting for covariates. Variables with *p* ≤ 0.05 in univariate analysis were entered into the model. Effect sizes were calculated using Cohen’s d. Values of 0.2, 0.5, and 0.8 represent small, medium, and large effect sizes, respectively. The confidence interval was set to 95% and the margin of error accepted was set to 5%. The *p* value was considered significant at *p* ≤ 0.05.

## 3. Results

In this study, we present our findings from a carefully structured evaluation involving 60 participants. These mothers, equally divided into a study group and a control group, were assessed for changes in psychological distress using the Depression Anxiety Stress Scale (DASS-21) and for enhancements in their problem-solving skills, pre- and post-intervention. Detailed demographic data including age, education, occupation, and the severity of the child’s autism were analyzed to contextualize the results. The efficacy of the Cognitive Behavioral Therapy (CBT) program is quantified through significant statistical measures, and the reliability of the tools used, such as (DASS-21) and Problem-Solving Inventory (PSI), is thoroughly examined. These results provide insightful conclusions on the intervention’s impact on reducing psychological distress and improving coping mechanisms among mothers of children with ASD.

Table 1 presents the demographic and baseline characteristics of the 60 study participants randomized into two groups: the study group (*n* = 30) and the control group (*n* = 30). The mean age of participants was 34.80 ± 10.25 years, with most aged between 20–40 years old. More than half of the participants (53.3%) had elementary education, 30% had secondary education and 16.7% had university education. Just under half were employed, while 53.3% were housewives. In terms of income adequacy, 56.7% reported inadequate monthly income. The children of the participants were predominantly male (68.4%), aged 4–12 years, with the severity of the autism spectrum disorder (ASD) symptoms relatively evenly distributed as mild (38.3%), moderate (40%), and severe (21.7%). Statistical analysis did not reveal significant differences between the study and control groups in terms of age, mother’s occupation, income adequacy, child’s age, child’s sex, and autism severity (all *p* > 0.05). However, there were significant differences in the level of education levels (*p* = 0.029) and marital status (*p* = 0.010), with the study group achieving higher levels of education and more likely to be married compared to the control group.

Table 2 presents the pre- and post-intervention scores on the Depression Anxiety Stress Scale (DASS-21) for the study group (*n* = 30) and control group (*n* = 30). At baseline, there were no significant differences between the two groups in depression, anxiety, stress or overall DASS-21 scores (*p* > 0.05). However, after the intervention, the study group showed statistically significant improvements compared to the control group in all DASS-21 measures (*p* < 0.05), including depression (mean score reduction of 2.53 vs. 0.4), anxiety (mean reduction of 5 vs. 1.54), stress (mean reduction of 3.93 vs. 0.47) and overall DASS-21 (mean reduction of 11.46 vs. 2.4). Within the study group, there were also significant reductions from pre- to post-intervention in the mean scores and percentages for depression, anxiety, stress, and overall DASS-21 (all *p* < 0.05). In contrast, the control group showed no significant pre- to post-intervention changes in any DASS-21 scores. Furthermore, the study group demonstrated significantly greater percentage reductions than the control group in depression (13.68% vs. 1.73%), anxiety (33.94% vs. 8.55%), stress (22.17% vs. 1.79%), and overall DASS-21 (24.75% vs. 3.86%) (all *p* < 0.05). In summary, these results indicate the intervention led to significant improvements in depression, anxiety, stress, and overall psychological distress in mothers of children with autism, compared to no significant changes seen in the control group. The reductions were statistically significant and clinically meaningful.

Table 3 shows a comparison of problem-solving skills between the study group (*n* = 30) and control group (*n* = 30) before and after the intervention. At baseline, there were no significant differences between the two groups in measures of problem-solving confidence, approach-avoidance style, personal control, or overall problem-solving skills (all *p* > 0.05). However, after the intervention, the study group demonstrated statistically significant improvements compared to the control group in all aspects of problem-solving skills assessed (all *p* < 0.05). Within the study group, there were also significant improvements from before to after the intervention in mean scores and percentages for confidence in problem solving, approach-avoidance style, personal control and general problem solving (all *p* < 0.05). In contrast, the control group did not show significant changes from pre-intervention to post-intervention. Additionally, the study group showed markedly higher percentage increases versus the control group in problem-solving confidence (8.21% vs. 1.45%), approach-avoidance style (13.28% vs. −2.24%), personal control (9.06% vs. 5.01%), and overall problem-solving skills (11.65% vs. 0.93%) (all *p* < 0.05). In summary, the intervention resulted in significant improvements in all measures of problem-solving ability in mothers of autistic children compared to no changes in the control group.

Table 4 presents results from multivariate linear regression models that examine the predictors of general psychological distress (DASS-21) and problem-solving skills (PSI) in mothers of children with autism spectrum disorder. Child age and autism severity were significant positive predictors of the overall DASS-21 score, explaining 74.1% of variance (*p* < 0.001). Specifically, older child age (B = 6.273, *p* = 0.005) and more severe autism symptoms (B = 2.268, *p* = 0.043) were associated with greater psychological distress in mothers. On the contrary, the only significant predictor of the overall PSI score was the adequacy of financial income (B = 4.938, *p* < 0.001), accounting for 75.8% of the variance (*p* < 0.001). Mothers with better financial adequacy reported better problem-solving abilities. However, the age and education of the mother, the marital status, and the age of the child were not significant predictors in either model. In summary, this regression analysis identified child factors (age and symptom severity) as predictors of maternal psychological distress, while financial resources predicted problem-solving skills. The models significantly explained over 70% of the variance in the outcomes.

## 4. Discussion

This randomized controlled trial provides novel evidence supporting the efficacy of a tailored cognitive behavioral therapy (CBT) program that emphasizes problem-solving skills training in reducing psychological distress among mothers of children with autism spectrum disorder (ASD). The findings align with and build upon prior research demonstrating the effectiveness of CBT in ameliorating distress and improving coping abilities in caregivers of children with disabilities [39,65,66,67]. However, this study makes a unique contribution by addressing the significant gap in evidence-based interventions tailored to the sociocultural context of the Arab world.

In line with our first hypothesis, mothers who participated in the 3-month CBT program exhibited statistically and clinically significant reductions in depression, anxiety, stress, and overall psychological distress compared to controls, as measured by the DASS-21. The large effect sizes signify that the improvements in emotional well-being were meaningful. As proposed by [68,69,70], these beneficial effects can be attributed to the cognitive restructuring techniques and emotional regulation skills taught in CBT, which enhanced self-awareness and adaptive thought patterns [39]. The supportive group environment likely also contributed to increased psychological comfort. Overall, equipping mothers with strategies to manage negative cognitions and emotions enabled them to better cope with the daily challenges of raising a child with ASD.

Additionally, the CBT group showed marked improvements across all domains of problem-solving skills assessed using the PSI, including confidence, approach-avoidance style, and personal control. This validates our second hypothesis that CBT would enhance problem-solving abilities, aligning with [71,72]. The cognitive and behavioral training helped mothers approach problems proactively, logically evaluate solutions, and feel empowered to implement coping strategies. These findings indicate that emphasizing practical problem-solving techniques within CBT can equip caregivers to handle difficulties more effectively, thereby boosting resilience.

Notably, regression analyses revealed that more severe ASD symptoms predicted higher maternal psychological distress, consistent with previous studies [73,74,75]. However, contrary to some past research [76], education level did not emerge as a predictor, suggesting the universal impact of ASD caregiving challenges across socioeconomic strata. Intriguingly, older maternal age predicted better post-intervention problem-solving, unlike some prior studies linking age and distress [77,78]. Additional research on age differences in coping abilities among these mothers may elucidate these mixed findings.

Future research should incorporate clinician and family-rated assessments, cultural factors, and evaluate maintenance of positive outcomes over an extended period post-intervention. Investigating potential indirect benefits to the children would also be worthwhile. Overall, this study provides solid preliminary evidence that should be built upon through further rigorous research.

In conclusion, these findings demonstrate that a tailored CBT intervention with dedicated problem-solving skills training can significantly improve the psychological well-being and coping capacities of mothers of children with ASD in the Arab world. This has important implications for implementing effective, culturally appropriate support programs for families impacted by ASD globally. The outcomes substantiate CBT’s versatility in ameliorating distress across diverse cultural settings. This study represents an important step towards filling the research gap and guiding mental health practice in the Arab world to enhance resilience in mothers of children with autism through evidence-based psychotherapeutic approaches.

### 4.1. Implications

The findings from this randomized controlled trial have several important implications for research and practice. Most notably, they provide evidence supporting the efficacy and cultural acceptability of a tailored CBT intervention emphasizing problem-solving skills training for mothers of children with ASD in the Arab world. This underscores the viability of implementing structured CBT programs as part of routine supportive services for families impacted by ASD in this region. The results also highlight the value of incorporating dedicated training in practical problem-solving techniques within CBT protocols to empower caregivers to better manage stressors. Additionally, the findings reveal a need for further research on predictors of maternal psychological distress and coping abilities to optimize treatment approaches. From a practice standpoint, they suggest that community mental health nurses can play a crucial role in delivering targeted CBT interventions to enhance resilience. Overall, by demonstrating the feasibility and benefits of a tailored CBT program in an Arab population, this study has implications for expanding access to effective psychotherapeutic approaches as standard supportive care for families affected by ASD globally.

### 4.2. Limitations

While this randomized controlled trial makes an important contribution to the literature, there are limitations that should be considered when interpreting the results, which include the following. (1) The small sample size (N = 60) from one geographic area limited the generalizability of the findings; studies with larger, more diverse samples are therefore needed. (2) The use of self-report measures for assessing psychological distress and problem-solving skills may have been subject to reporting bias; additional evaluation by clinicians would strengthen the outcomes. (3) The short follow-up period means that the sustainability of positive effects over a longer term is unclear. Future research should incorporate longer-term follow-up assessments as well as addressing the impact of social issues. (4) There was a lack of evaluation of potential indirect benefits on the children with ASD. It is therefore unknown if improving maternal mental health translated to outcomes for the children. (5) The control group received minimal contact, which could have influenced responses. A control group with an active placebo treatment would help to account for nonspecific effects. (6) There were also significant differences in the level of education and marital status between groups, which could have impacted coping abilities independent of the intervention. (7) The single-blinded study design means that participants knew their group allocation, which could have biased the responses. Double blinding would have been preferable. (8) A possible Hawthorne effect may have occurred, where participants change behaviors due to an awareness of being studied; however, use of a control group helped to account for this. (9) The findings may not generalize to fathers or other primary caregivers of children with ASD. Research on broader populations is therefore warranted. (10) The study did not evaluate which specific components of the multifaceted CBT intervention were most effective. Future component analysis would therefore be beneficial.

In summary, while this study provides valuable preliminary insights, the limitations highlight the need for further robust research to substantiate and extend the findings.

## Figures and Tables

**Figure 1 behavsci-14-00046-f001:**
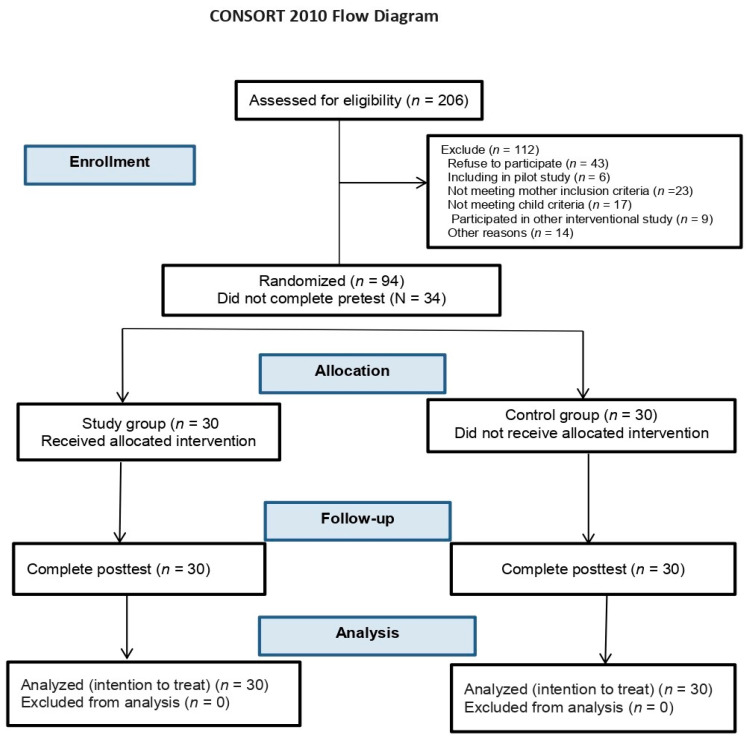
Flow chart.

**Table 1 behavsci-14-00046-t001:** Demographic and baseline characteristics of study participants by group.

Participants’ Characteristics	All Participants (*n* = 60)		Study Group (*n* = 30)		Control Group (*n* = 30)		χ^2^	*p*
	No.	%	No.	%	No.	%		
Age (years)								
20–30	25	41.7	11	36.7	14	46.7	0.789	0.674
30–40	14	23.3	7	23.3	7	23.3		
40–50	21	35.0	12	40.0	9	30.0		
Mean ± SD	34.80 ± 10.25		35.53 ± 10.81		34.07 ± 9.78		0.551	0.584
Education								
Elementary education	32	53.3	11	36.7	21	70.0	7.081 *	0.029 *
Secondary education	18	30.0	13	43.3	5	16.7		
University	10	16.7	6	20.0	4	13.3		
Mother occupation								
Working	28	46.7	15	50.0	13	43.3	0.268	0.605
Housewife	32	53.3	15	50.0	17	56.7		
Adequacy of monthly income								
Adequacy	26	43.3	11	36.7	15	50.0	1.086	0.297
Inadequate	34	56.7	19	63.3	15	50.0		
Marital status								
Married	32	53.3	21	70.0	11	36.7	6.696 *	0.010 *
Other	28	46.7	9	30.0	19	63.3		
Child’s age								
4–8	32	53.3	18	60.0	14	46.7	1.071	0.301
8–12	28	46.7	12	40.0	16	53.3		
Child’s sex								
Male	41	68.4	25	61	16	39	0.601	0.438
Female	19	31.6	9	47.3	10	52.7		
Severity of autism								
Mild	23	38.3	12	40.0	11	36.7	0.287	0.866
Moderate	24	40.0	11	36.7	13	43.3		
Severe	13	21.7	7	23.3	6	20.0		

χ^2^: Chi-square test. * Indicates statistically significant difference between groups (*p* < 0.05).

**Table 2 behavsci-14-00046-t002:** Pre- and post-intervention DASS-21 scores in study and control groups.

Depression Anxiety Stress Scale (DASS-21)	All Participants (*n* = 60)				Study Group (*n* = 30)				Control Group (*n* = 30)				Test of Sig. (p1)	Test of Sig. (p2)
	Pre		Post		Pre		Post		Pre		Post			
	No	%	No	%	No	%	No	%	No	%	No	%		
Depression														
Normal	2	3.3	11	18.3	2	6.7	9	30.0	0	0.0	2	6.7	χ^2^ = 2.263 (MCp = 0.386)	χ^2^ = 8.909 * (0.012 *)
Mild	36	60.0	38	63.3	16	53.3	19	63.3	20	66.7	19	63.3		
Moderate	22	36.7	11	18.3	12	40.0	2	6.7	10	33.3	9	30.0		
Total score	13.70 ± 3.10		12.23 ± 2.90		13.60 ± 3.50		11.07 ± 2.27		13.80 ± 2.70		13.40 ± 3.02		t1 = 0.248 (0.805)	t1 = 3.378 * (0.001 *)
% score	32.62 ± 7.38		29.13 ± 6.91		32.38 ± 8.33		26.35 ± 5.41		32.86 ± 6.42		31.90 ± 7.20			
t2 (p0)	3.496 * (0.001 *)				4.080 * (<0.001 *)				0.797 (0.432)					
% reduction	7.70 ± 24.80				13.68 ± 29.11				1.73 ± 18.16				U = 270.0 * (0.004 *)	
Anxiety														
Normal	3	5.0	9	15.0	2	6.7	7	23.3	1	3.3	2	6.7	χ^2^ = 0.927 (MCp = 1.000)	χ^2^ = 20.076 * (MCp < 0.001 *)
Mild	11	18.3	15	25	5	16.6	13	43.3	6	20	2	6.6		
Moderate	23	38.3	26	43.3	12	40.0	8	26.7	11	36.7	18	60.0		
Severe	23	38.3	10	16.7	11	36.7	2	6.7	12	40.0	8	26.7		
Total score	14.40 ± 4.40		11.13 ± 5.11		14.13 ± 4.55		9.13 ± 5.22		14.67 ±4.31		13.13 ± 4.19		t1 = 0.466 (0.643)	t1 = 3.274 * (0.002 *)
% score	34.29 ± 10.48		26.51 ± 12.16		33.65 ± 10.83		21.75 ± 12.42		34.92 ± 10.26		31.27 ± 9.98			
t2 (p0)	5.195 * (<0.001 *)				5.732 * (<0.001 *)				1.916 (0.065)					
% reduction	21.03 ± 34.61				33.94 ± 28.79				8.55 ± 35.61				U = 237.000 * (0.003 *)	
Stress														
Normal	22	36.7	29	48.3	13	43.3	22	73.3	9	30.0	7	23.3	χ^2^ = 1.304 (MCp = 0.788)	χ^2^ = 19.377 * (MCp < 0.001 *)
Mild	13	21.7	14	23.3	6	20.0	6	20.0	7	23.3	8	26.7		
Moderate	20	33.3	9	15.0	9	30.0	2	6.7	11	36.7	7	23.3		
Severe	5	8.3	8	13.3	2	6.7	0	0.0	3	10.0	8	26.7		
Total score	17.83 ± 5.0		15.63 ± 6.77		16.80 ± 5.29		12.87 ± 4.75		18.87 ± 4.54		18.40 ± 7.42		t1 = 1.623 (0.110)	t1 = 3.423 * (0.001 *)
% score	42.46 ± 11.90		37.28 ± 16.07		40.0 ± 12.61		30.74 ± 11.20		44.92 ± 10.81		43.81 ± 17.66			
t2 (p0)	2.676 * (0.010 *)				5.894 * (<0.001 *)				0.325 (0.747)					
% reduction	10.19 ± 39.71				22.17 ± 20.95				1.79 ± 49.73				U = 315.500 * (0.046 *)	
Overall DASS-21														
Total score	45.93 ± 9.03		39.0 ± 11.32		44.53 ± 9.91		33.07 ± 8.59		47.33 ± 7.97		44.93 ± 10.67		t1 = 1.206 (0.233)	t1 = 4.744 * (<0.001 *)
% score	36.46 ± 7.16		30.95 ± 8.98		35.34 ± 7.86		26.24 ± 6.82		37.57 ± 6.33		35.66 ± 8.47			
t2 (p0)	5.456 * (<0.001 *)				10.656 * (<0.001 *)				1.201 (0.240)					
% reduction	14.31 ± 22.72				24.75 ± 14.07				3.86 ± 25.04				U = 188.000 * (<0.001 *)	

χ^2^: Chi-square test. MC: Monte Carlo. t1: Student *t*-test. t2: paired *t*-test. U: Mann–Whitney test. p1: *p* value for comparing between the studied groups in pre. p2: *p* value for comparing between the studied groups in post. p0: *p* value for comparing between pre and post in each group. *: Statistically significant at *p* ≤ 0.05.

**Table 3 behavsci-14-00046-t003:** Comparison of problem-solving skills between study and control groups.

Problem-Solving Skills Scale	All Participants (*n* = 60)		Study Group (*n* = 30)		Control Group (*n* = 30)		Test of Sig. (p1)	Test of Sig. (p2)
	Pre	Post	Pre	Post	Pre	Post		
Problem-Solving Confidence (PSC)								
Total score	40.23 ± 5.24	38.23 ± 5.35	40.73 ± 5.01	36.83 ± 5.11	39.73 ± 5.50	39.63 ± 5.30	t1 = 0.737 (0.464)	t1 = 2.084 * (0.042 *)
% score	53.15 ± 9.52	49.52 ± 9.73	54.06 ± 9.11	46.97 ± 9.28	52.24 ± 9.99	52.06 ± 9.64		
t2 (p0)	2.204 * (0.031 *)		3.158 * (0.004 *)		0.080 (0.937)			
% reduction	3.38 ± 18.22		8.21 ± 17.34		1.45 ± 18.06		t = 2.113 * (0.039 *)	
Approach–Avoidance Style (AAS)1								
Total score	57.00 ± 5.49	52.35 ± 6.54	56.80 ± 5.83	48.90 ± 4.82	57.20 ± 5.22	55.80 ± 6.25	t1 = 0.280 (0.780)	t1 = 4.944 * (<0.001 *)
% score	51.25 ± 6.86	45.50 ± 8.10	51.00 ± 7.28	41.13 ± 6.03	51.50 ± 6.53	49.87 ± 7.58		
t2 (p0)	5.357 * (<0.001 *)		6.816 * (<0.001 *)		1.368 (0.182)			
% reduction	7.76 ± 11.11		13.28 ± 10.21		−2.24 ± 9.14		t = 4.409 * (<0.001 *)	
Personal Control (PC)								
Total score	19.32 ± 3.41	18.17 ± 3.89	19.03 ± 2.55	16.87 ± 4.61	19.60 ± 4.12	19.47 ± 2.47	t1 = 0.640 (0.525)	t1 = 2.723 * (0.009 *)
% score	57.27 ± 13.65	52.67 ± 15.57	56.13 ± 10.21	47.47 ± 18.43	58.40 ± 16.50	57.87 ± 9.90		
t2 (p0)	5.220 * (<0.001 *)		2.078 * (0.047 *)		0.188 (0.852)			
% reduction	2.02 ± 31.16		9.06 ± 28.95		5.01 ± 32.16		t = 1.781 (0.080)	
Overall problem-solving skills scale								
Total score	116.55 ± 8.58	108.77 ± 9.48	116.57 ± 7.46	102.60 ± 6.75	116.53 ± 9.70	114.93 ± 7.65	t1 = 0.015 (0.988)	t1 = 6.623 * (<0.001 *)
% score	52.84 ± 5.36	47.98 ± 5.92	52.85 ± 4.66	44.13 ± 4.22	52.83 ± 6.06	51.83 ± 4.78		
t2 (p0)	5.290 * (<0.001 *)		7.691 * (<0.001 *)		0.944(0.353)			
% reduction	6.29 ± 9.38		11.65 ± 7.75		0.93 ± 7.71		t = 5.370 * (<0.001 *)	

t1: Student *t*-test. t2: paired *t*-test. p1: *p* value for comparing between the studied groups in pre. p2: *p* value for comparing between the studied groups in post. p0: *p* value for comparing between pre and post in each group. *: Statistically significant at *p* ≤ 0.05.

**Table 4 behavsci-14-00046-t004:** Predictors of psychological distress and problem-solving skills in mothers of children with autism.

	Overall DASS-21						Overall PSI					
	B	Beta	t	*p*	95% CI		B	Beta	t	*p*	95% CI	
					LL	UL					LL	UL
Age of mother	−0.023	−0.036	0.284	0.779	−0.189	0.144	0.188	0.482	3.908 *	0.001 *	0.088	0.287
Education	−1.337	−0.146	1.060	0.300	−3.945	1.272	0.871	0.154	1.156	0.260	−0.687	2.429
Adequacy of financial	3.402	0.244	1.793	0.086	−0.522	7.325	4.938	0.574	4.359 *	<0.001 *	2.595	7.281
Marital status	−1.163	−0.079	0.588	0.562	−5.255	2.930	−0.208	−0.023	0.176	0.862	−2.652	2.236
Child age	6.273	0.458	3.128 *	0.005 *	2.125	10.420	−0.229	−0.027	0.191	0.850	−2.706	2.248
Severity of autism	2.268	0.263	2.144 *	0.043 *	0.080	4.456	0.702	0.132	1.111	0.278	−0.605	2.008
R2 = 0.741, F = 10.944 *, *p* < 0.001 *							R2 = 0.758, F = 12.016 *, *p* < 0.001 *					

F, *p*: f and *p* values for the mode. R2: coefficient of determination. B: unstandardized coefficients. Beta: standardized coefficients. t: *t*-test of significance. CI: confidence interval. LL: lower limit. UL: upper limit. *: Statistically significant at *p* ≤ 0.05.

## Data Availability

The data are available upon reasonable request.

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
