# Peer review of "Effectiveness of Cognitive Behavioral Therapy (CBT) on Psychological Distress among Mothers of Children with Autism Spectrum Disorder: The Role of Problem-Solving Appraisal"

_behavsci, 2024, doi:10.3390/bs14010046_

Round 1

Reviewer 1 Report

Comments and Suggestions for Authors

Excellent study and contribution to the literature. Thank you for the opportunity to review. There are a few minor things that should be addressed prior to publication, but otherwise very strong paper. Here are the two minor items below:

-Remove “(give reasons)” from Analysis portion of Figure 1 Flow chart.

-Level of education and marital status significant differences between the Study and Control groups should be added to the limitations and/or mentioned in the Discussion. The Study group having a higher percentages of marriage and education levels could have contributed to their coping strategies and problem solving abilities. Each could be important components of resilience to stress.

Reviewer 2 Report

Comments and Suggestions for Authors

First and foremost, I'd like to thank the authors for putting together this manuscript. It was good to read, interesting to think about, and genuinely kept me engaged not just as a scientist, but as a reader too. I think that the authors have very strong organizational skills and do a good job of making their empirical arguments in an efficient manner. I've not heard of CBT with an emphasis on problem-solving used to address maternal distress among parents of children with ASD diagnoses, so this was refreshing new material. 

As it stands, I am recommending that this paper is published with minor revisions. I think that the overall strength of this paper's arguments, while it could be improved, is adequate for publication, and the justification for the study's publication is sufficient. I think that your methodology is sufficient as is, but would like more detail in a few key areas. My specific comments are as follows.

Section 2.5.3 and 2.5.2: Why exclude "severe" levels of maternal distress and participants with co-occurring ASD and ID? I recognize that many studies which are looking to test new methodologies do so in order to remove these factors as potentially confounding. However, as you have stated, these methods have a history of prior empirical support, although not with Arab populations specifically; parents of these children could have even formed a third group if there were concerns about them reacting differently to the CBT service. I think a short explanation of this would be helpful, or more detailed itemization as a limitation.

2.9.2: CBT sessions are described as lasting 60-90 minutes. However, an additional 30 minutes of CBT can greatly change the course of a session. What was the cause of this variation? Were some groups simply longer-lasting than others, did session length fluctuate from week to week, etc.? A mean run time and standard deviation measure, as well as an answer to the above question, would be most welcome. 

2.9.4: Similar to the above comment, how long was this "final segment" of each section for discussion? My assumption is that this was the reasoning behind the variability of the CBT session length, but I would like this to be more clearly stated. 

4.1: I've noticed that you have two descriptions of study limitations, one towards the end of section 4.1, and the other as the section 4.3. Unless another reviewer is requiring inclusion of the limitations section in 4.1, remove it; it is redundant, and your description in 4.3 is much more comprehensive. I would also consider changing the formatting of 4.3's limitations to a textual, rather than bullet-point, font, for increased coherence with the rest of the text, even if each limitation was individually numbered within the text (e.g., 1) small sample size, 2) use of self-report....). 

Given that you have collected a large amount of self-report data, why did you not collect social validity data as well? My apologies if I overlooked this data, but it would be very nice to have; if you do not have it, this would be good to include as a limitation.

Your Figure 1 also needs some visual work. The data within the figure is excellent, I've produced multiple figures like this and as far as I am concerned it is well-constructed. However, boxes are regularly off-center, lines clip into text boxes, and the figure contains inconsistent text font and size. The image overall is blurry as well. I've attached a PDF that I recently received from the Journal of Applied Behavior Analysis on saving images in a higher-content way; doing so will make your Figure 1 look more professional.

As a last couple of things, you seem to be missing a "5" header (the headers jump from 4.3 to 6), and your APA references section needs just a bit of formatting work; in particular, journal article titles are typically in lowercase. 

Overall, this was a good paper to read. I'm particularly grateful that you used prospective registration, and the justification for your study is strong. I hope that you will be able to publish this source soon!

Reviewer 3 Report

Comments and Suggestions for Authors

Absctract

Suggested improvements:

·         Consider specifying the exact duration and frequency of the CBT sessions to provide clearer details on the intervention's structure.

·         Provide more context or rationale behind the selection of specific problem-solving appraisal techniques incorporated into the CBT program.

·         Include a brief overview of the sample characteristics (e.g., demographics) to offer a better understanding of the participants involved in the study.

The introduction provides a comprehensive overview of Autism Spectrum Disorder (ASD) and its impact on caregivers, particularly mothers, setting the context for the study. It effectively outlines the prevalence of ASD, the challenges faced by caregivers, and the psychological distress experienced by mothers, emphasizing the need for effective interventions. The integration of cognitive behavioral therapy (CBT) and problem-solving techniques is well-detailed as a potential solution.

However, there are areas where the introduction can be improved:

·         Specificity: While the introduction broadly highlights the challenges faced by mothers of children with ASD, it could benefit from more specific examples or cases to illustrate the psychological distress and societal challenges experienced. This would provide a deeper understanding of the issues at hand.

·         Research Gap: The study identifies a research gap regarding the application of CBT among mothers in the Arab world. However, it would strengthen the introduction to specify why this gap exists and how the cultural context might influence the effectiveness of interventions like CBT. Incorporating examples or existing literature that emphasize these cultural differences would enhance the argument.

·         Clarity on Study Objectives: While the study aims and hypotheses are outlined towards the end of the introduction, they could be introduced earlier to provide a clearer roadmap for readers, allowing them to understand the study's objectives from the outset.

·         Integration of Nurse Role: The role of community mental health nurses is highlighted towards the end of the introduction, but their specific role in the study or the intervention could be articulated more explicitly, especially in connection to the implementation of CBT and problem-solving techniques.

·         Transition and Flow: The introduction covers a wide range of information, sometimes without clear transitions. Reorganizing or breaking down some sections into smaller, more digestible portions could improve the flow and readability.

·         Conclusion of the Introduction: The conclusion briefly reiterates the need for research in the field but could be more explicit in summarizing the objectives and potential contributions of the study.

Overall, while the introduction provides a solid background and rationale for the study, enhancing specificity, linking cultural context more explicitly, and refining the study objectives could strengthen its impact and clarity.

The "Materials and Methods" section provides a detailed description of the study design, including research questions, hypotheses, participant selection criteria, interventions, and data collection methods. Here's a critical evaluation along with suggestions for improvement:

Strengths:

·         Clear Research Questions and Hypotheses: The research question and hypotheses are well-defined, outlining the specific aim of the study and the expected outcomes regarding psychological distress and problem-solving skills.

·         Detailed Design and Intervention Description: The section offers a comprehensive explanation of the randomized controlled trial (RCT) design, intervention procedures, and the control group's activities, providing a clear contrast for the study's effectiveness evaluation.

Suggestions for Improvement:

·         Justification for Sample Size: While the study mentions a power analysis, providing a brief justification for choosing the sample size based on prior effect sizes or expected outcomes from similar studies could enhance the rationale.

·         Detailed Participant Recruitment: While the study mentions recruitment from three specific locations, explaining why these sites were chosen and if they represent a diverse or specific population would strengthen the sampling strategy.

·         Participant Blinding Explanation: Clarifying the blinding or masking procedures for both participants and investigators after randomization could enhance the validity of the trial.

·         Detailed Intervention Content: While the intervention methods are broadly outlined, providing a more granular breakdown of each session's content and structure, possibly in a tabular or sequential format, would offer a clearer understanding of the intervention's components.

·         Assessment of Intervention Fidelity: Ensuring and reporting fidelity to the intervention is critical. Adding details on how the fidelity of the CBT program was monitored or assessed would enhance the study's robustness.

·         Statistical Analysis Plan Clarity: The section detailing statistical analyses is comprehensive but might benefit from a brief summary at the beginning to provide an overview before delving into specific tests and methodologies.

Overall, while the "Materials and Methods" section is comprehensive, adding more specific details, justifications, and transparency regarding procedures and analyses could further strengthen the study's validity and reproducibility.

The text presents a scientific study assessing the impact of Cognitive Behavioral Therapy (CBT) on mothers of children with Autism Spectrum Disorder (ASD), focusing on psychological distress and problem-solving skills. The study involved 60 participants split equally into a study group and a control group. Here's a critical evaluation and suggestions for improvement:

Strengths:

·         Structured Approach: The study design involving a controlled group and the detailed assessment of demographic data shows a meticulous approach.

·         Use of Standardized Measures: The use of validated tools like DASS-21 and PSI enhances the study's reliability and comparability.

·         Statistical Analysis: Statistical measures and significance levels are well-documented, providing clarity on the obtained results.

·         Clear Findings: Results indicate significant improvements in psychological distress and problem-solving skills in the intervention group compared to the control, which is a substantial finding.

Areas for Improvement:

·         Clarity in Presentation: Some sections seem a bit convoluted due to the dense presentation of statistical data. Breaking down complex statistical data into concise points or creating clear visual representations like charts might aid readability.

Language Enhancement Suggestions:

·         Consistency in Terminology: Maintain consistency in terminology and abbreviations throughout the text. For instance, in some places, "ASD" is spelled out while in others it's abbreviated.

·         Clarity in Descriptions: Some sentences could benefit from clearer phrasing for ease of understanding.

Overall, the study offers valuable insights into the effectiveness of CBT in improving the mental well-being of mothers with autistic children. Enhancing the clarity and presentation could further strengthen the paper's impact.

The discussion section of this scientific article offers a comprehensive analysis of the findings while acknowledging the study's strengths and limitations. Here's a critical evaluation considering its nature as a scientific article:

Strengths:

·         Clarity of Results: The discussion effectively outlines the positive impact of tailored cognitive behavioral therapy (CBT) on reducing psychological distress among mothers of children with autism spectrum disorder (ASD).

·         Building on Prior Research: The authors skillfully connect their findings with existing research, demonstrating how this study augments previous knowledge in the field of CBT for caregivers.

·         Cultural Context Sensitivity: Highlighting the significance of addressing sociocultural contexts in interventions is commendable, especially regarding the Arab world, indicating a broader applicability of CBT interventions.

·         Quantitative Analysis: The discussion provides statistical evidence, citing effect sizes and measured improvements, enhancing the credibility of the findings.

Suggestions for Improvement:

·         Greater Depth in Limitations: While the section on limitations is comprehensive, a deeper exploration of certain constraints, especially regarding potential biases and methodological shortcomings, would be beneficial.

·         Detailed Explanation of Methodology: It would be advantageous to elaborate on the methodologies employed, explaining the selection of measures, controls, and the rationale behind the design choices.

·         Further Discussion on Implications: Expanding on the implications beyond the immediate application in the Arab world could enhance the article's relevance to a broader audience.

Language and Structure:

The language and structure of the discussion are suitable for a scientific article. However, for greater clarity, some sections could benefit from breaking down complex sentences or technical terms to make the content more accessible to a wider audience.

Conclusion:

Overall, the discussion is well-structured, systematically presenting the study's findings, implications, and limitations. To enhance its impact, deeper exploration of limitations, a broader discussion on implications, and a more detailed explanation of methodologies could be considered.

This article makes a valuable contribution to understanding the efficacy of tailored CBT in addressing psychological distress among caregivers in the Arab world. With some refinement, it could significantly influence future research and interventions in this area.
